# Automated quality assessment of cognitive behavioral therapy sessions through highly contextualized language representations

**Nikolaos Flemotomos**[1]*, **Victor R. Martinez**[1], **Zhuohao Chen**[1], **Torrey A. Creed**[2], **David C. Atkins**[3], **Shrikanth Narayanan**[1]

**1** Signal Analysis and Interpretation Lab, University of Southern California, Los Angeles, CA, United States of America, **2** Department of Psychiatry, University of Pennsylvania, Philadelphia, PA, United States of America, **3** Department of Psychiatry and Behavioral Sciences, University of Washington, Seattle, WA, United States of America

* flemotom@usc.edu

**Data Availability Statement:** Data cannot be shared publicly due to the confidential nature of

## Abstract

During a psychotherapy session, the counselor typically adopts techniques which are codified along specific dimensions (e.g., 'displays warmth and confidence', or 'attempts to set up collaboration') to facilitate the evaluation of the session. Those constructs, traditionally scored by trained human raters, reflect the complex nature of psychotherapy and highly depend on the context of the interaction. Recent advances in deep contextualized language models offer an avenue for accurate in-domain linguistic representations which can lead to robust recognition and scoring of such psychotherapy-relevant behavioral constructs, and support quality assurance and supervision. In this work, we propose a BERT-based model for automatic behavioral scoring of a specific type of psychotherapy, called Cognitive Behavioral Therapy (CBT), where prior work is limited to frequency-based language features and/or short text excerpts which do not capture the unique elements involved in a spontaneous long conversational interaction. The model focuses on the classification of therapy sessions with respect to the overall score achieved on the widely-used Cognitive Therapy Rating Scale (CTRS), but is trained in a multi-task manner in order to achieve higher interpretability. BERT-based representations are further augmented with available therapy metadata, providing relevant non-linguistic context and leading to consistent performance improvements. We train and evaluate our models on a set of 1,118 real-world therapy sessions, recorded and automatically transcribed. Our best model achieves an $F_1$ score equal to 72.61% on the binary classification task of low vs. high total CTRS.

## Introduction

Psychotherapy is an intervention based on verbal interaction between patients and trained professionals, aimed at treating mental health and behavioral disorders. The effectiveness of psychotherapy is widely accepted [1, 2] with millions of people seeking professional help at a yearly basis [3]. Cognitive Behavioral Therapy (CBT) [4] is a particular type of

recorded psychotherapy sessions as well as the original consent forms. These data are recordings and transcripts of psychotherapy sessions, and therefore contain potentially sensitive and identifying information that would cause harm to participants if it were shared. The University of Pennsylvania ethics committee has imposed restrictions on the sharing of these data. Specifically, these data were shared through a Data Use Agreement that restricted their use specifically for this study but does not allow the data to be shared publicly. For more information, please contact the University of Pennsylvania's IRB at 215.573.2540 or PROVOST-IRB@pobox.upenn. edu

**Funding:** TAC, DCA, and SN received a grant from the National Institutes of Mental Health (NIMH; https://urldefense.com/v3/__http://www. nimh.nih.gov__;!!Llr3w8kk_Xxm! 8sJoTAeo6YcHAzWqdhdhnGyO3- cJbW0rUDZXtQ71aLrZJ6e75bwsFFf_fiRLYvo$) - grant number R56 MH118550. DCA and SN received a grant from the National Institute of Alcoholism and Alcohol Abuse (NIAAA; https://urldefense.com/v3/__http://www. niaaa.nih.gov__;!!Llr3w8kk_Xxm! 8sJoTAeo6YcHAzWqdhdhnGyO3- cJbW0rUDZXtQ71aLrZJ6e75bwsFFf_uQY5hjl$) - grant number R01 AA018673. The funders had no role in study design, data collection and analysis, decision to publish, or preparation of the manuscript.

psychotherapy that aims at shifting the patient's patterns of thinking by changing maladaptive cognitions and beliefs connected to behavioral problems. It is one of the most popular psychotherapeutic approaches [5] with strong evidence connecting its methods with positive clinical outcomes [6].

Given its wide popularity and applicability to a variety of mental health problems, performance-based measures that ensure high quality of CBT provision are deemed essential [7]. The gold-standard method for monitoring therapy quality is *behavioral coding* [8], a process during which trained coders listen to audio recordings and read session transcripts (recently, text-only psychotherapy interactions are also being adopted to complement spoken conversations [9]) in order to assess specific therapeutic skills. For CBT, in particular, the most widely used coding scheme is the Cognitive Therapy Rating Scale (CTRS), that defines a set of 11 session-level codes reflecting skills and techniques specific to the intervention (the CTRS manual is available from https://www.academyofct.org/general/custom.asp?page=CTRSCRRS [Young JE, Beck JS; Unpublished]). This approach, however, is prohibitive for scale-up to widespread practice in real-world clinical settings due to time and cost demands, which means that the vast majority of CBT sessions are simply not evaluated.

Recent technological advances have given rise to a digital healthcare era with numerous applications focusing on mental health [10]. Automatic behavioral coding, in particular, has drawn a lot of interest over the last few years (e.g., [11–15]) and holds promise for more efficient training, more effective supervision, and more positive clinical outcomes. However, despite being one of the most dominant psychotherapy interventions, the literature focusing on computational analysis and evaluation of CBT sessions is relatively scarce, partly because of limited available data. The existing proposed systems mainly depend on frequency-based and hand-crafted linguistic features [16, 17], or study CBT-related constructs appearing in short text excerpts which are not part of an actual therapy session [18]. CBT sessions, however, are at least 20-30 minutes long (sometimes even longer than an hour), with a typical session consisting of several tens or hundreds of talk turns and utterances. At the same time, the behavioral constructs under examination reflect complex structural, conceptual, and communicative aspects of the therapy that the existing approaches potentially fail to capture.

Inspired by the recent success stories of large pre-trained context-rich language models across numerous Natural Language Processing (NLP) tasks [19–21], in this work we adapt a BERT model [19] to the domain and use it for the downstream task of CBT quality assessment. The metric we are focusing on is the total CTRS score which is equal to the sum of the 11 individual codes and is used in clinical practice to evaluate a practitioner's degree of competence in delivering CBT [22]. We train systems which either directly classify a session with respect to the total CTRS or model all the constituent codes in a multi-task approach, thus enhancing interpretability. Side information from available metadata is also included to the final models, leading to improved predictive power of the systems, compared to only using linguistic cues.

To the best of our knowledge, this is the largest study focusing on automated evaluation of cognitive behavioral therapy, employing more than 1,100 recorded and automatically transcribed CBT sessions with hundreds of different patients and providers. Scaling up performance-based CBT evaluation to real-world usage opens up exciting opportunities towards the provision of fast and cost-effective feedback to the practitioner. Such feedback can be beneficial for training new therapists or for maintaining acquired CBT-related skills, and can eventually lead to improved mental health care services and more positive outcomes for the patients.

## Materials and methods

### Datasets

The Beck Community Initiative (BCI) at the University of Pennsylvania provides high-quality psychotherapy training to community clinics and, through this work, has generated a large archive of recorded CBT sessions [7], many of which are accompanied by CTRS scores. Therapists who have participated in the program are primarily female (75.7%) and identify as White (49.2%), Black or African American (36.3%), Asian (3.7%), Native American or Alaskan Native (1.0%), Native Hawaiian or Pacific Islander (0.2%), or other (9.9%). Additionally, a subset of therapists (13.1%) identify their ethnicity as Hispanic or Latino.

BCI has implemented CBT in more than 75 diverse community mental health contexts and the sessions included in the current study are intentionally heterogeneous in terms of population, presenting problem, and setting, in order to support the generalizability of the findings. Rather than limiting inclusion to sessions with patients with a specific disorder, BCI uses a transdiagnostic CBT approach that is appropriate across the broad spectrum of presenting problems and populations seen in community care. The CTRS is the established gold-standard measure of competence in CBT and is the most commonly used measure to evaluate transdiagnostic CBT skills.

Out of the available sessions, 292 have been sent for professional transcription. The selection of the particular sessions was done so that there is fair representation of sessions across the entire range of the total CTRS scale and the audio quality is above a certain threshold. Specifically, we estimated the overall Signal-to-Noise Ratio (SNR) and set a minimum threshold equal to 7dB. We used those human-transcribed sessions to adapt and evaluate an automatic speech transcription pipeline, which was later used to transcribe a total of 1,118 CBT sessions (including the 292 already mentioned). This number comes from the initial pool of available sessions after excluding those marked as non-English and the ones for which not all the 11 CTRS codes were available. The dataset mostly comprises dyadic conversations (i.e., one therapist and one patient) but also contains some sessions with multiple speakers (e.g., group therapy).

The transcription pipeline is based on [23], after domain adaptation to the CBT data, and consists of voice activity detection, diarization, speech recognition, and speaker role recognition (i.e., therapist vs. patient). Consecutive utterances assigned to the same role with a silence gap shorter than two seconds are merged together. The role recognition module operates at the speaker level, which means that for each speaker identified by the diarization algorithm, we estimate whether it is more probable to be a therapist or a patient. Since we always have a single therapist per session, we also tried a version of the speaker recognition module where only one speaker (from the ones identified by the diarization algorithm) is assigned the therapist role and the rest are considered patients. However, initial experimentation showed that this approach led to poor performance because of increased missed therapist speech.

Adaptation was based on 100 transcribed sessions, leaving 1,018 CBT sessions for further experimentation. The final Word Error Rate (WER), when the pipeline is evaluated on the remaining 192 human-transcribed sessions, is 45.81%, with the therapist-attributed error being 41.19%. Even though the automated transcription WER is high, those numbers are inflated since they are highly affected by fillers (e.g., 'um', 'huh', etc.) and other idiosyncrasies of conversational speech.

Each of the 11 CTRS codes listed in Table 1 is scored by a trained human coder on a 7-point Likert scale (0-6). For the dataset used in this study, a total of 28 doctoral-level CBT experts served as raters, with regular reliability meetings held among them to prevent rater drift. Giving equal importance to all the 11 CTRS dimensions, CBT researchers take into

**Table 1. The 11 CBT quality codes defined by CTRS.**

| abbreviation | meaning |
| --- | --- |
| ag | agenda |
| fb | feedback |
| un | understanding |
| ip | interpersonal effectiveness |
| co | collaboration |
| pt | pacing and efficient use of time |
| gd | guided discovery |
| cb | focusing on key cognitions & behaviors |
| sc | strategy for change |
| at | application of techniques |
| hw | homework |

consideration the total CTRS which is the sum of the 11 components. In clinical practice, a total CTRS above or equal to 40 indicates competent delivery of CBT, whereas a score less than 40 could suggest, for example, that additional training is required for the particular practitioner [24]. The focus of this work is on the binary classification problem of the total CTRS (below/above 40). The distribution of all the CTRS codes is given in Fig 1. Even though the total CTRS follows an approximately normal distribution, after binarization the problem is unbalanced with the dataset becoming skewed towards the class with non-competent CBT delivery (total CTRS below 40), which represents 76.23% of the sessions in our dataset.

The goal of the current study is to examine the feasibility of applying an automated system for the task of CTRS prediction, based solely on linguistic information. We should note, here, that even though all the skills defined by CTRS can be demonstrated, at least partially, through verbal cues, some of them are expected to be less amenable to purely language-based representations. As suggested by the results in [16], the human-centric codes associated with establishing a good relationship with the patient, namely the CTRS dimensions of understanding,

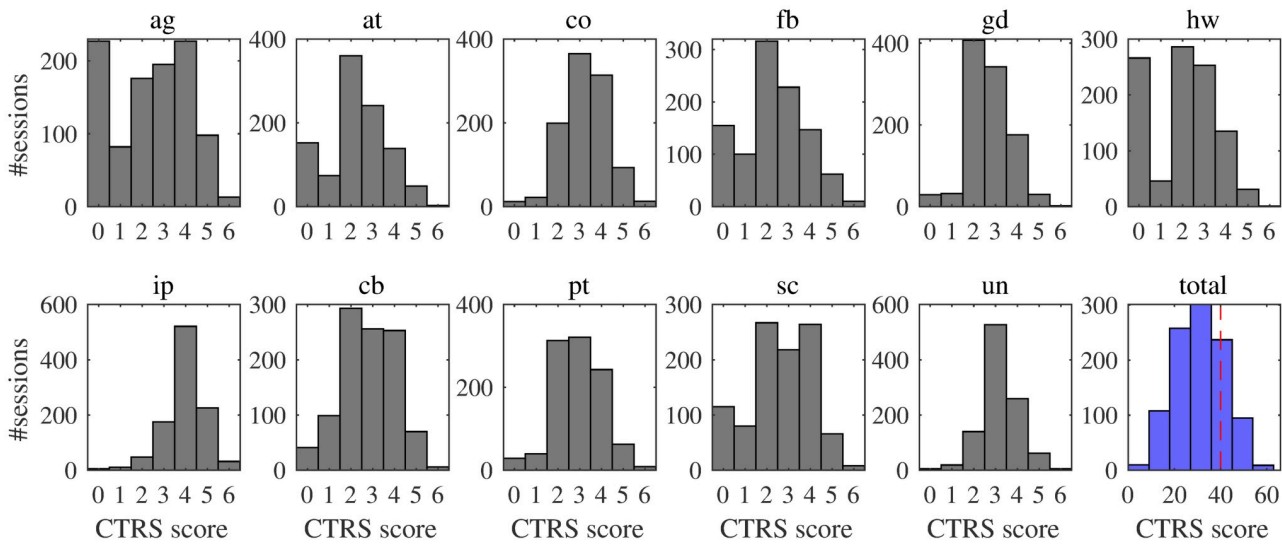

**Fig 1. Distribution of the 11 CTRS codes (and the total CTRS) across the Likert scale.** A total CTRS score above or equal to 40 indicates competent delivery of CBT.

**Table 2. Examples of utterances within sessions with high/low scores for the CTRS dimensions of agenda and homework.**

| code | score | utterance |
|------|-------|-----------|
| agenda | high | [. . .] Maybe we should put that on your *agenda* for today. Is that what you want to talk about? |
| | low | So what are we going to talk about today? What's the *agenda* for today? |
| homework | high | [. . .] I know we haven't seen each other in a couple weeks. But some of the *homework* that was given, [. . .] |
| | low | Because I always took time and sat down with them with *homework* and everything [. . .] |

A score is considered low if it is below 4 and high if it is above or equal to 4 on the 0-6 Likert scale. As shown in those examples (drawn from the manually transcribed sessions), merely the usage of certain words or phrases is not always indicative of high competence in corresponding CBT skills. *agenda*: in the low-scored session, the therapist tried to set an agenda with this utterance, but failed to follow it throughout the entire session. *homework*: in the low-scored session, the word 'homework' is used, but refers to kids' coursework and not to any CBT-related assignment.

interpersonal effectiveness, and collaboration, are especially difficult to be modeled through language features. Even for some of the codes for which simple linguistic observations, such as the choice of words [16], can give valuable insights towards competent delivery of CBT, the task is often more challenging than it may seem on the surface. For instance, as shown by the examples given in Table 2, it is crucial to take into consideration the entire history of the conversation and not just rely on linguistic cues found within isolated utterances.

The task is, of course, becoming even more challenging, given the fact that the linguistic information used in this work comes from a fully automated transcription pipeline with a relatively high error rate, which results to error propagation across the various sub-modules and inevitably affects the downstream task of behavioral coding. Previous works, however, have shown that, for the specific problem, the performance degradation of such language-based systems because of inaccurate transcriptions is relatively small [17].

For all the sessions, a limited amount of metadata are also available. In particular, the variables taken into consideration in this work are:

1. the *clinic* where the session took place: The dataset consists of sessions delivered by 383 therapists across 25 clinics. Different agencies may follow different standards and practices, which may be partly indicative of the quality of the delivered services.

2. the *level of care*, characterizing the intensity of treatment services: The sessions are clustered into 6 distinct categories along this dimension; namely, inpatient, outpatient, intensive outpatient, residential, school-based, and assertive community treatment [25].

3. the *population* for which the services are intended: Treatment is often tailored to a specific audience, so the type of clients served may be indicative of a therapist's behavioral skills. Our dataset consists of sessions clustered into 9 population groups, defined by age ("child", "adolescent", "adult", "geriatric"), presenting problem ("substance use", "serious mental illness"), or other group-specific characteristics ("LGBTQI", "forensic", "homelessness").

4. the *assessment time* with respect to when the CBT-focused training of the corresponding therapist took place: Each therapist participating in the study attends a workshop organized by BCI to receive CBT training. Couselors' CBT skills are expected to improve after participation in CBT training; they are also expected to keep improving as they practice CBT and receive expert consultation and feedback. Thus, the availability of such information can be useful for the task of CTRS prediction. There is a total of 7 timestamps characterizing a

session along this dimension (e.g., pre-workshop, post-workshop, three months after workshop ends, etc.).

Apart from the aforementioned CBT data, we additionally employ a set of 4,268 recorded psychotherapy sessions automatically transcribed from a university counseling center [23] which will be used for domain adaptation. We will denote this as the UCC set. Those sessions span a wide range of psychotherapy approaches (including, but not limited to, CBT) and have not been coded following the CTRS. Despite the expected differences between the two sets (e.g., the UCC sessions are focused on concerns common among college students, such as anxiety, whereas the CBT data span a wider range of mental health problems), several common linguistic patterns in psychotherapy are expected to be shared. So, this set is considered suitable to adapt the BERT model which will be used to extract linguistic representations of the CBT sessions. An additional advantage of using the UCC sessions for adaptation is that they have been transcribed using the same transcription pipeline used for the CBT dataset, so BERT is expected to be adapted not only to the psychotherapy domain, but also to common transcription-specific errors. The size of the two datasets in terms of duration and number of utterances/words is provided in Table 3.

We note that the current study is an analysis of archival data. CBT data were initially collected and used as part of a training program. Participants had signed a written consent to record and to have the audio provided for educational and training purposes. The University of Pennsylvania Institutional Review Board approved the usage of the de-identified, archival program eval data for this study (Protocol #827045, granted in March 2017). Additionally, the University of Utah Institutional Review Board approved the usage of the UCC data (Protocol #83132, granted in March 2017). For the initial UCC data collection, participants had signed a written consent to record and to have the audio provided for research purposes.

## Single-task approach

The current work is focused on the binary classification problem of low vs. high (lower than 40 vs. greater than or equal to 40) total CTRS score, which is of particular interest from a clinical perspective [7, 22]. Thus, it is natural, as a first step, to build a model viewing the problem as a single task where the output is exactly a binary variable denoting whether the therapist is considered to have successfully adhered to CBT-related skills or not. Since all the codes defined by CTRS only depend on therapist behavior and are not directly related to the patient (e.g., 'did the therapist set a clear *agenda* for the session?'), we can focus on the utterances (talk turns) assigned to the former. Within this framework, we propose the architecture illustrated in Fig 2.

**Table 3. Size of the datasets used to train and evaluate the proposed models.**

| dataset | #sessions | #therapists | #talk turns | session duration [min] (mean ± std) | turns per session (mean ± std) | words per turn (mean ± std) |
|---|---|---|---|---|---|---|
| CBT-all | 1,018 | 383 | 438,830 | 35.5 ± 12.8 | 431.1 ± 229.3 | 12.7 ± 24.3 |
| CBT-onlyT | | | 206,457 | 17.7 ± 8.6 | 202.8 ± 96.2 | 14.2 ± 24.1 |
| UCC-all | 4,268 | 59 | 1,873,126 | 38.9 ± 10.0 | 438.9 ± 200.1 | 15.4 ± 29.6 |
| UCC-onlyT | | | 896,879 | 15.1 ± 7.4 | 210.1 ± 106.8 | 12.7 ± 23.9 |

The statistics are given both when *all* the utterances and only the ones assigned to the therapist (*onlyT*) are taken into account. We consider as a talk turn any segment of the session where a single speaker is present and there is a maximum silence gap of 2 seconds between consecutive words. The session duration is estimated as the sum of all the turn durations, based on the timestamps predicted by the rich speech transcription pipeline.

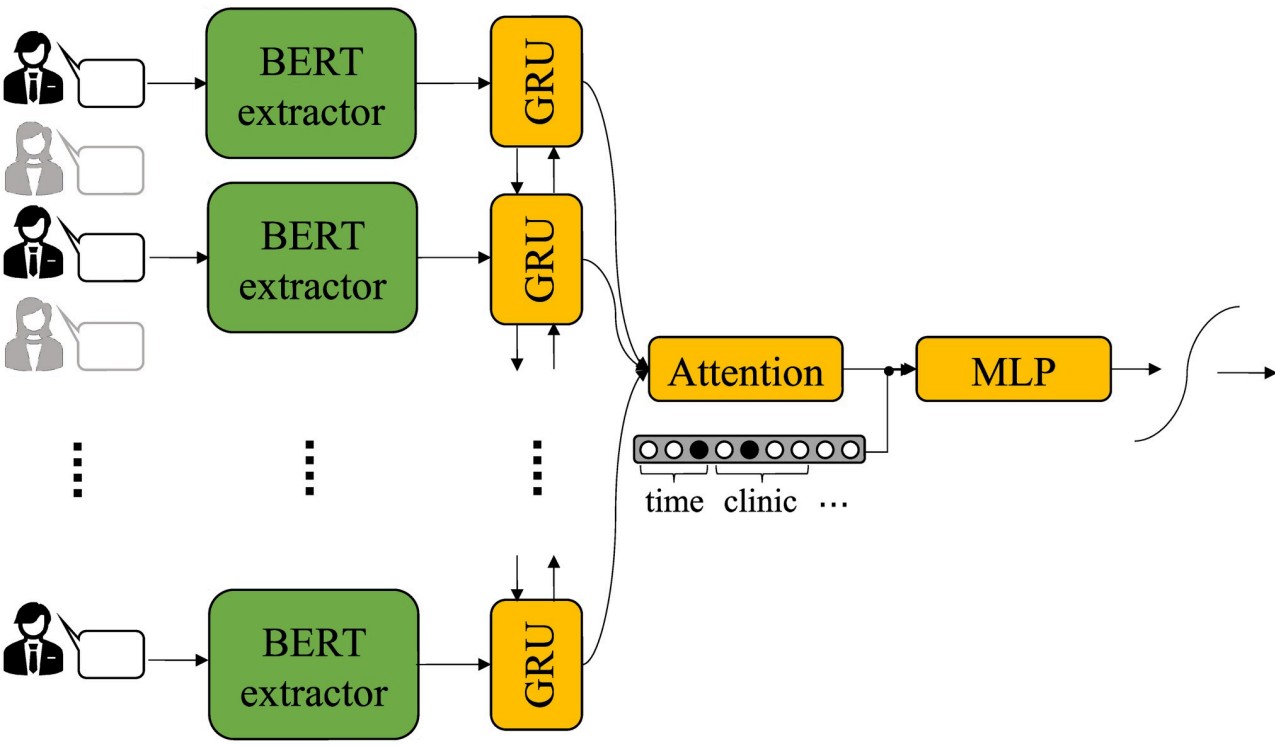

**Fig 2. Proposed architecture for total CTRS score classification following a single-task approach.** Here, only the utterances attributed to one of the interlocutors (i.e., therapist) are used.

Systems based on BERT representations have achieved remarkable performance across various text classification tasks [26], while a large number of existing studies report that domain-specific BERT variants outperform the generic model [27–30]. Based on such evidence, we first adapt a generic pre-trained BERT model to the psychotherapy domain by continuing training on an external, related/in-domain dataset (in our case, the UCC data). This adapted model can then be used to extract fixed-dimensional linguistic representations for each available utterance.

The sequence of utterance representations forms the input to a recurrent neural network with attention, an architecture commonly used to solve a wide range of NLP problems [31–33]. In more detail, the sequence of BERT representations is fed to a bidirectional recurrent layer, built with Gated Recurrent Units (GRUs) [34], that accordingly outputs a sequence of hidden vectors, each one of which takes into consideration not only the corresponding utterance but the entire context of the session (i.e., what is said before and after the utterance). The hidden vector corresponding to each recurrent cell is considered to be the concatenation of the forward and backward outputs of the specific cell. This sequence is fed to an additive self-attention layer [35] which models the entire session as a weighted average of the information encoded in the hidden vectors. That way, the system learns which parts of the session are useful in order to construct a meaningful session representation with respect to the final task of overall CTRS prediction. This representation can now be concatenated with the available session-level metadata information, represented by one-hot encoded variables [36]. Finally, a sigmoid non-linearity, applied after a Multilayer Perceptron (MLP), gives the desired output.

The network is optimized based on the binary cross-entropy loss function. In order to deal with class imbalance, we weight each sample using class weights inversely proportional to class frequencies.

## Multi-task approach

The single-task architecture described in the previous section does not take into account what exactly the total CTRS represents. However, the total CTRS score is estimated as the unweighted sum of the 11 individual CTRS codes and different codes typically represent completely different CBT skills. Those skills are related to specific linguistic patterns and are often applied by the therapist during different parts of the session. For example, the therapist is expected to set an appropriate *agenda* towards the beginning of the session that includes the specific topics the patient would like to discuss in that session. Similarly, an important aspect of a successful CBT session is incorporating *homework* relative to the therapy. That includes reviewing previous homework (typically done towards the beginning of the session) and assigning new homework for the coming week (typically done towards the end of the session). Finally, there are codes which reflect communicative skills expected to be displayed through-out the entire session. For instance, the therapist is expected to communicate both an empathic *understanding* and a clinical understanding based on a cognitive conceptualization to the patient through appropriate verbal responses.

In order to implicitly incorporate such knowledge in the network, we propose applying multi-task learning, a paradigm that can often achieve more robust and generalizable representations, leading to more powerful models [37]. Our proposed multi-task approach is depicted in Fig 3 and, as shown, the first steps (BERT-based feature extraction and

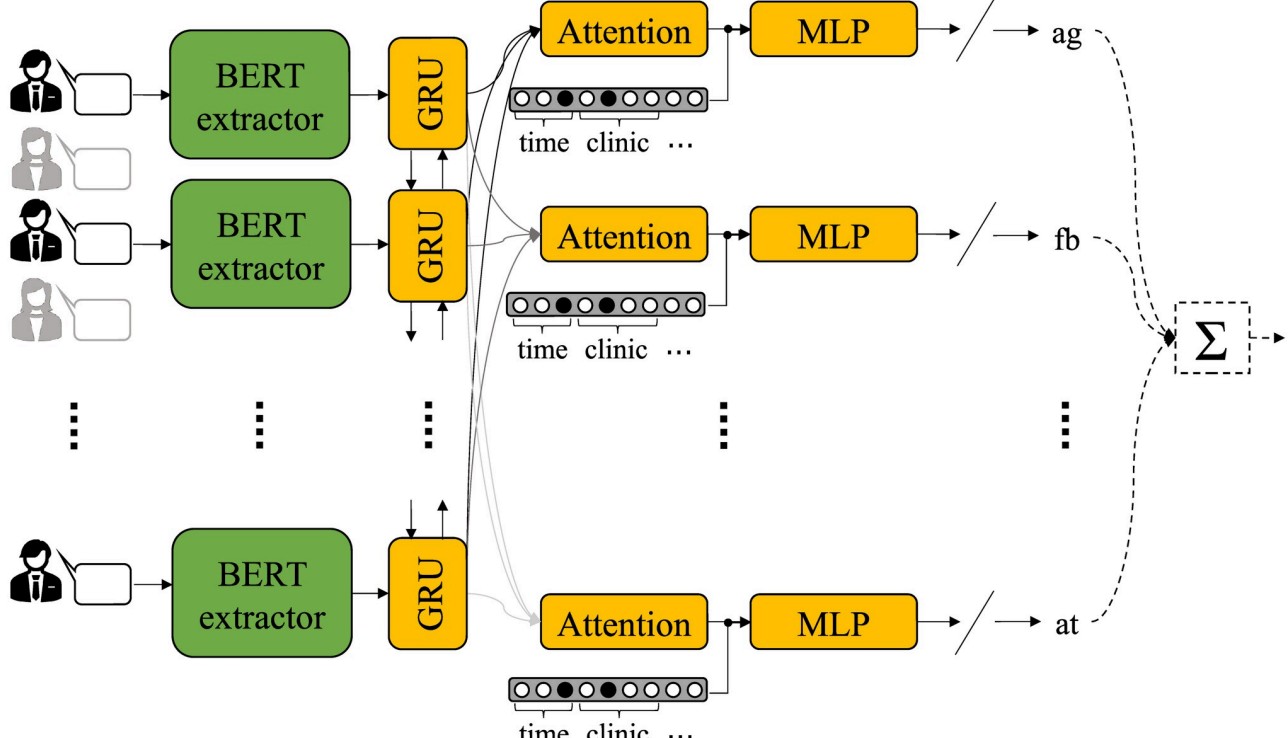

**Fig 3. Proposed architecture for total CTRS score classification following a multi-task approach modeling each CTRS code.** Here, only the utterances attributed to one of the interlocutors (i.e., therapist) are used.

bidirectional recurrency) are the same as in the single-task approach described previously. However, instead of directly modeling the total CTRS score, the system now tries to separately model each one of the 11 codes, with each code defining a "task" for the network.

The sequence of hidden vectors from the recurrent layer is shared across all the tasks and is the input to 11 different attention layers, each one associated with a specific CTRS dimension. That way, the network can attend to what is important for the prediction of a particular code. As previously, metadata information in the form of one-hot encoded variables is concatenated to the context vector learned by the attention layers and is passed through MLPs with linear (instead of sigmoid) output activation functions. So, a continuous output—restricted in the range [0, 6]—represents each CTRS code. Those codes can later be added and the binarized sum corresponds to the desired classification outcome (low vs. high total CTRS). The loss function to be optimized during training is $L = \sum_{i=1}^{11} L_i$, where $L_i$ is the mean squared error associated with the $i$-th code. We underline that during training the network learns to predict scores for all the 11 CTRS dimensions in a multi-task fashion, as described, and the total CTRS is only estimated as their sum at a later phase for evaluation purposes.

An additional advantage we get following this approach is enhanced interpretability, an aspect of crucial importance for systems used in real-world clinical settings. Instead of viewing the overall system as a black box giving information about the total CTRS score, we can now track specific attributes which led to the classification of a session as competent or not. That way, if such a system is used to provide feedback to counselors, this can be targeted to specific areas in need of improvement.

## Experimental setup

**BERT adaptation.** For this work we employ the pre-trained uncased BERT-base model (available at https://github.com/google-research/bert) as a feature extractor, which provides 768-dimensional embeddings. BERT embeddings are extracted at the utterance level by average-pooling the last layer. Initial experimentation showed that, for our problem, average pooling yields better results than using the [CLS] token, which is typically used in classification tasks [19, 26]. The pre-trained model is tuned by continuing training on the set of 4,268 UCC sessions (Table 3) after a random 90% − 10% train-eval split at the session level. Two adapted BERT models are built and evaluated: i) a model adapted on all the available utterances, called *psychBERT*, and ii) a model adapted only on the therapist-attributed utterances, called *therapistBERT*. In both cases, a maximum utterance length of 64 tokens is assumed and tuning takes place for 10,000 steps with a learning rate equal to $2 \cdot 10^{-5}$ and minibatch size equal to 64. We note that only 1.9% of the total number of utterances in the CBT set were longer than 64 tokens and had to be cropped. Since we expect a lot of short utterances during evaluation, we create training sequences shorter than the maximum length with a relatively high probability (equal to 0.5). Following the official recommendations, we apply a masking rate equal to 15% [19].

**Network design.** We apply both proposed architectures (single-task and multi-task) for the downstream task of psychotherapy quality evaluation based on the total CTRS score. In order to study whether non-therapist language contains information useful to the task, we run experiments both using all the available utterances and only the therapist-attributed ones. For each case, we additionally examine the effectiveness of the available metadata.

The models are built and trained using Tensorflow [38]. We use a single bidirectional recurrent layer composed of GRU cells with 64 hidden units each. The attention weights are learned through dense layers (one for each attention mechanism) of 10 hidden units. The classification MLPs (Figs 2 and 3) comprise one hidden dense layer with 20 units and ReLU

activations and one output dense layer with either sigmoid (for the signle-task approach) or linear (for the multi-task approach) activation.

The Adam optimizer [39] is employed, with initial learning rate equal to 0.001. The models are trained for a maximum of 200 epochs with early stopping based on validation loss, with patience set equal to 10 epochs. When focusing only on therapist-attributed utterances, the maximum sequence length (session length) is set to 256 utterances and a batch size equal to 128 is used. When all the utterances are taken into consideration, the maximum sequence length is set to 512 utterances and the batch size is decreased to 64 in order to meet the limitations set by the available computational resources (one NVIDIA GeForce GTX 1080 Ti).

## Results and discussion

### BERT model performance

We first evaluate the BERT model, before and after adaptation, on the next sentence prediction task. Next sentence prediction teaches BERT to learn long-term dependencies across sentences and can give useful insights for long documents, such as therapy transcripts. Long contextual information is especially important for our final task of CTRS prediction, since the therapist is expected to demonstrate relevant skills throughout the entire session. Additionally, relations between consecutive utterances/sentences might indicate linguistic patterns and techniques often encountered in psychotherapy, such as open or closed questions and reflective listening. Such techniques are widely adopted in other psychotherapeutic approaches, but are often incorporated in CBT as well [40], while there are results supporting that they are useful as a supplemental information for predicting CTRS scores [17].

The accuracy results of the model, when evaluated on the CBT sessions, are given in Table 4. As shown, adaptation leads to substantial improvements, for both psychBERT and therapistBERT. The large performance gap when BERT-base is used comes at no surprise: The higher accuracy for the task when all the utterances are taken into consideration (compared to the case when only the therapist utterances are evaluated) is due to the fact that the base model can more accurately represent naturalistic conversations (e.g., questions-answers), compared to predicting the next utterance of a specific person, skipping one of the interlocutors. However, after adaptation, the system does an almost equally good job for the two cases (and even slightly better when only applied to the therapist utterances).

Comparing the results between the second and third row of Table 4 is of high interest, since we can see that there is a very small degradation—possibly smaller than expected—when we use therapistBERT to evaluate on all the available utterances, or, similarly, when we use psychBERT to evaluate only on the therapist-attributed ones. From those results, it seems that the tuning process mainly adapts the BERT model on psychotherapy-specific language and on general characteristics of conversational speech (e.g., high frequency of short sentences and fillers), which are common both when we take patient-attributed utterances into consideration and when we do not. Of course, this behavior is partly due to the fact that even when adapting only on the therapist segments of the UCC data (therapistBERT), it is inevitable that some

**Table 4. Accuracy (%) for the next sentence prediction task before and after BERT adaptation when the models are evaluated on the CBT dataset.**

| model | CBT set all utterances | CBT set therapist utterances |
|---|---|---|
| BERT-base | 60.03 | 40.00 |
| psychBERT | **69.53** | 71.08 |
| therapistBERT | 69.18 | **71.66** |

**Table 5. $F_1$ score (%) based on 10-fold cross validation.**

| utterance representation | metadata info | all utterances | | therapist-only utterances | |
|---|---|---|---|---|---|
| | | single-task | multi-task | single-task | multi-task |
| BERT-base | ✗ | 63.43 | 61.03 | 63.88 | 62.40 |
| | ✓ | 65.42 | 70.13* | 66.80† | 71.25* |
| adapted BERT (psychBERT/therapistBERT) | ✗ | 64.10 | 62.04 | 65.52 | 63.76 |
| | ✓ | 66.94† | 71.56* | 68.52* | **72.61** * |

The adapted BERT is psychBERT when all the utterances are used and therapistBERT when only the ones assigned to the therapist are used. Paired bootstrap tests were performed with respect to BERT-base, singe-task approach ($n = 10^5$).

† $p < 0.05$,

* $p < 0.01$

patient-attributed utterances are encountered, too, because of potential errors in the speaker diarization and role recognition modules of the rich transcription pipeline [23]. Similarly, during psychBERT training, consecutive talk turns do not necessarily belong to different speakers, so consecutive sentences used for the "next sentence prediction" task often belong to the same speaker (e.g., therapist). In any case, for the following experiments on CTRS prediction, we always use psychBERT when taking all the utterances into account and therapistBERT when only using the therapist-attributed ones, since those are the models that give the best results, even if by a small margin.

## CTRS prediction

The experimental results using our proposed models are given in Table 5. All the results reported are based on a 10-fold cross validation scheme so that there is no therapist overlap between the folds (the patient IDs are not known). The evaluation metric used is the macro-averaged $F_1$ score. Additionally, Table 6 provides the relative improvements to the overall performance when applying each one of the 4 proposed techniques: i) adapting BERT to the domain, ii) providing metadata information, iii) following the multi-task approach, and iv) taking only the utterances assigned to the therapist into consideration.

For completeness, we also trained and evaluated a linear support vector machine with unigram-based tf-idf features (selecting the 32 best features based on a univariate $F$-test), an approach that gave the best results in [16]. This yielded an $F_1$ score equal to 66.58% when using all the utterances and 67.73% when using only the utterances assigned to the therapist. Even though our language-only models did not beat this frequency-based baseline, the results do get better when we provide the available metadata information, especially coupled with the multi-task approach. This demonstrates the validity of our proposed techniques to enhance

**Table 6. Contribution of each proposed technique to the performance of the system.**

| proposed technique | no | yes | relative improvement |
|---|---|---|---|
| BERT adaptation | 65.54 | 66.88 | +2.04% |
| metadata info | 63.27 | 69.15 | +9.29% |
| multi-task | 65.58 | 66.85 | +1.94% |
| only therapist | 65.58 | 66.84 | +1.92% |

Each row gives the mean $F_1$ score (%) across all the remaining $2^3 = 8$ combinations when the corresponding technique is (*yes*) or is not (*no*) applied.

the predictive power of the models, but, at the same time, leaves room for further experimentation and improvement regarding the linguistic representations used.

Comparing the results of Table 5 in more detail, it is apparent that the inclusion of patient utterances does not provide complementary information for the task of total CTRS prediction. Even though psychotherapy is a conversational interaction and one would assume that the entire history of the dialogue could improve the predictive power of the system, we should take into account that CTRS codes are focused only on therapist behavior. The results support the initial hypothesis that focusing on therapist-only language is sufficient, and tends to be robust, to assess therapist-related behaviors within the proposed framework. Additionally, it is shown that, as expected, the adapted BERT extractor yields better linguistic representations than the base model, at least with respect to our final goal.

Overall, the best results are achieved when we use the multi-task approach after providing the available metadata information. However, it is interesting to note that, while metadata information is consistently beneficial to the system, it is not always clear whether the multi-task approach leads to improved results, especially when metadata information is not provided. Given the availability of such side information, though, the multi-task architecture does indeed boost the overall performance. This is likely due to the fact that, in this case, metadata improve the robustness while estimating each one of the codes, thus improving the overall robustness.

It should be highlighted, here, that, while non-linguistic side information proves to be highly beneficial for the particular dataset, further investigation is required to study which variables are expected to be readily available in the general case of CBT quality assessment in the real world. For instance, even though the assessment time with respect to CBT training appears to be a reasonable proxy of CBT quality, should we expect that such information be always provided to the system? In a real-world scenario, such decisions could actually be informative of how an interface used in clinical settings should be built, i.e., what therapist-related information should be asked for during a new user registration.

Finally, it is important to note that, most of the time, the value of the used metadata variables was known to the human annotators. So, it is not clear at this point whether the performance boost is due to actual useful complementary information that such non-linguistic variables carry or due to modeling annotator bias. For example, annotators may be biased towards specific clinics because of exceptional reputation, or they may tend to score therapists lower when they are in earlier stages of training as compared to those who are completing training. However, we should highlight that, for the specific dataset, all the annotators were required to demonstrate calibration prior to coding [7]. During that process, all coders were blind to the metadata information other than the one coder in the group who was the assigned coder for the official rating. Strong inter-rater reliability results (ICC = 0.84) suggest that raters may not be influenced by having access to such external information.

## Localization of CTRS codes

CBT is a highly structured psychotherapeutic approach and this is reflected in several of the CTRS codes used to assess the quality of a session. Using the attention mechanisms of our proposed architecture, we can identify salient utterances for the task of CTRS prediction and, through this process, we can reveal the aforementioned structure. In other words, since the network attends to specific parts of the session for different codes, we can examine how the practitioner focuses on different aspects of CBT throughout therapy.

In Fig 4 we illustrate how the attention weights progress, on average, through time for three exemplar codes: *agenda*, *homework*, and *feedback*. The therapist typically sets an appropriate

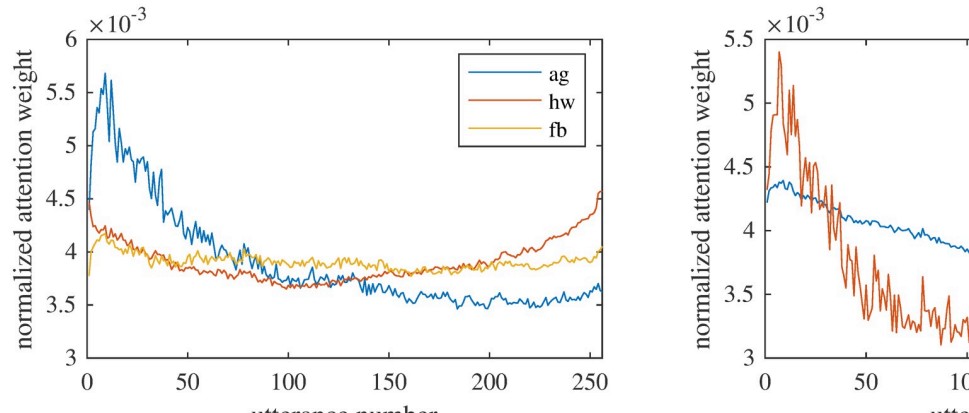

**Fig 4. Mean attention weights across all the sessions, combining the results from all the testing folds of the cross validation.** (left) Weights for the codes *agenda*, *homework* and *feedback* from the multi-task architecture. (right) Weights for the *total CTRS* from the single-task architecture and *mean* weights of all the 11 individual codes from the multi-task architecture.

agenda towards the beginning of the session establishing key items to be discussed and addressed. Even though they are expected to follow the agenda throughout the session, and thus merely setting an agenda is not sufficient, it seems that, for the task of automatic coding within the proposed framework, there is a big spike of salience towards the beginning of the session which is gradually diminished. Similarly, for the CTRS dimension of homework, the network tends to attend more to the beginning and the end of the session, where the counselor typically reviews and assigns cognitive therapy homework, respectively. Finally, feedback is typically elicited from the patient through appropriate questions asked frequently throughout the session and, thus, there are not expected localized patterns. Indeed, the attention mechanism of the system seems to give, on average, approximately equal weights to different utterances during the session. Of course, this is an aggregate behavior and it does not mean that specific patterns and saliency spikes are not observed in individual sessions.

In general, we observed that for some codes the system attends more to the beginning and/or the end of the session and for some of them the saliency curves are relatively flat—aggregate spikes in the middle of the session were not observed for any code. The saliency curve generated from the attention weights of the single-task approach for the total CTRS (Fig 4) is also in accordance with that observation and follows a similar pattern with the one generated as the unweighted average of the 11 individual CTRS codes.

## Ethical and practical implications

When dealing with such sensitive topics, like psychotherapy and automatic evaluation of one's job performance, it is important to step back and reflect on the implications of our work. Speech and language processing models keep getting better at an unprecedented pace, and the same is expected for the downstream tasks that those models are used for. But which are the key areas we should focus on when using those models and when developing techniques that exploit people's data and affect users' lives? At least four questions need to be answered with respect to the specific application we are dealing with in this work:

1. *How can we protect patients' sensitive data?*
   A necessary prerequisite of being able to automatically assess the quality of a psychotherapy session is having access to a collection of real-world data. Computational models can then be trained on such data in order to discover underlying patterns. There is no doubt,

however, that psychotherapy sessions contain extremely sensitive information, since patients often build a trust bond with their therapist, unbosom themselves, and disclose thoughts and secrets they are afraid or ashamed to share even with friends and family. Thus, it is of utmost importance that such data be treated with extreme caution during all the phases of a study, including data collection, storage, and usage.

The privacy of all the participating individuals—both patients and therapists—should be respected and they should all give their consent to be recorded, while also having the right to withdraw and stop the recording at any time during therapy. Data should then be carefully stored to dedicated machines with restricted access. Researchers should honor the trust that the patient has placed in the counselor and make every effort to prevent third parties from accessing and misusing those data.

Of course, the current study is governed by restrictions imposed by the relevant Institutional Review Board (IRB), while all the data used for research purposes were de-identified. However, IRB-related restrictions should only provide a formal framework and constitute just a first step. Treating psychotherapy data cautiously should be an ethical obligation of each individual researcher involved and not merely the result of external restrictions imposed to them [41].

2. *What if such a system is used to blindly evaluate a therapist?*

   Since CTRS provides quantifiable metrics for quality assessment, the proposed system can be used to evaluate a therapist's performance and their adherence to specific psychotherapeutic skills. Scoring competence and providing performance-based feedback to counselors is an important part of training and leads to improved quality of CBT services [42]. BCI, in particular, has been providing feedback to clinicians based on CTRS scores for the last 14 years and, along with offering technical support and intensive consultation, is successfully infusing CBT techniques into mental health services addressing a wide range of disorders [7].

   In a dystopian scenario, however, machine-based feedback could mean therapists losing their jobs because of not meeting minimum standards set by a black-box model and students being disappointed because of getting "low scores" from an automated system. It should be made clear that our goal is not to replace human supervision, but rather augment the supervisor's efficiency and additionally offer a tool for self-assessment. Moreover, it is important that the users be adequately trained to understand the meaning of automatically generated evaluation scores. This is why the focus should be on highly interpretable models. One of such are the attention-based systems which can give insights into specific salient parts and patterns of the inputs. The attention mechanisms employed in our proposed system can reveal the structure of a CBT session and point to particular segments where some of the underlying dimensions of the total CTRS score are in potential need of improvement.

3. *What if the system is wrong? Are there any explicit additional requirements before using such a system in clinical settings?*

   It is important for a practical realistic system to incorporate confidence metrics and quality safeguards. The described system depends on a series of machine learning models where things can simply go wrong. Establishing confidence metrics for the quality of the automatic transcription (e.g., speech recognition—induced errors) and the final CTRS prediction (e.g., applying thresholds on the final sigmoid non-linearity) would enhance the transparency of the models and would help practitioners trust them and introduce them into the clinical world.

While it is crucial that the system can inform the user about potential errors, it is equally important that the user be able to question model predictions when they disagree with the conclusions [43]. This may include, for example, the practitioner manually changing an erroneous transcription or indicating a specific segment of the session where they believe they exhibited a particular CBT-related skill (e.g., providing a time window when they assigned cognitive therapy homework). Such a strategy would not only enhance user engagement and willingness to adopt the system, but would also provide useful input data which can lead to ongoing model adaptations and improvements.

4. *How can we ensure that the computational systems employed are not affected by various biases introduced by the training data?*
   Machine learning systems are only as good as the training data with which they are provided. For example, it is known that state-of-the-art NLP systems trained on mainstream available data perform poorly when applied to language of authors from minority groups [44]. At the same time, people's perceptions about psychotherapy differ across different cultures and any psychotherapeutic approach, including CBT, needs to be culturally adapted [45]. Thus, an automated system for psychotherapy quality assessment also needs to be adapted to the actual use case and it is essential that the final user be aware of the training conditions and the potential limitations which are due to condition mismatch. A system used in real-world settings which is systematically skewed against dyads of counselors-patients who belong to minority groups would only promote a culturally homogeneous mental health care system and would exacerbate existing disparities in mental health treatment among different racial and ethnic groups [46]. In an effort to address this issue, the dataset used in the current study is based on a population that has rich representation of people from historically marginalized groups.
   On a similar note, human annotators almost never reach perfect agreement [47] and, as a result, models trained on data manually annotated by a small pool of coders may suffer from annotator biases and not generalize well [48]. In order to alleviate the problem and build a reliable system, a large and ideally diverse pool of human coders should be employed. Additionally, information which can potentially affect coders' objectivity and is not crucial to the annotation process (e.g., assessment time with respect to CBT training) should probably be not available to them at annotation time. However, something like that is not always possible, or even desirable, given the cost associated with creating a large in-domain dataset and the complex nature of psychotherapy. In our case, the available therapy sessions have been labeled by a large number of raters who did have access to metadata information. The ultimate goal of behavioral coding is the provision of quality feedback about skills and competence to counselors. In order to do so, human supervisors need to know about details related to the course of treatment, including information about previous sessions, in order to better assess an individual session. Thus, the decision of allowing annotators to have access to such information is essentially a trade-off decision between objectivity and external validity.

## Prior work

### Highly contextualized language models

Large pre-trained language models have lately led to several developments and breakthroughs in numerous NLP tasks, including text classification, text generation, question-answering, and natural language inference. Those language models are usually built based on the concept of

Transformer [49]. Using several stacked Transformer blocks, systems like GPT [50] and BERT [19] were able to push the limits of NLP.

BERT opened up a new era in NLP with several variants having been proposed, which are usually targeted at specific tasks and applications, or address certain BERT limitations. In its original form, for instance, BERT is only able to handle relatively short pieces of text. Doc-BERT [51] was proposed to address this limitation by focusing on the task of document classification. Psychotherapy code prediction can be viewed as a variant of document classification, with the "document", however, being a dialogue. ToD-BERT [52] has been specifically proposed to incorporate the power of BERT in modeling task-oriented dialogues. Similarly, DialoGPT [53] builds upon GPT-2 [54] focusing on dialogues, but for the task of response generation.

Domain-specific BERT variants have been also developed for particular fields which use, for example, specialized vocabulary (e.g., [27, 28]). In the clinical domain, the authors in [29] adapted the BERT embeddings both on general clinical corpora and on discharge summaries in particular. Similarly, the BERT model was adapted on clinical notes for the task of hospital readmission prediction in [30]. However, those adaptation processes are based on written text and do not focus on medical conversations, such as psychotherapy. In this work, we tuned BERT embeddings specifically to address the linguistic idiosyncrasies found within a psychotherapy session and we showed that our system can achieve improved results when compared to the non-adapted model.

## Incorporating metadata information

Conditioning linguistic representations on external side information, often distilled from expert knowledge or available metadata, is beneficial to numerous NLP tasks. For example, movie genre has been used for the task of violence rating prediction from movie scripts [55], while external note-level and patient-level attributes have been used to classify clinical notes in electronic health records [56]. The authors in [36] proposed and compared several methods for incorporating such external knowledge as contextual information in recurrent language models. Their results suggest that feeding both the linguistic and auxiliary data to an extra dense layer after the recurrent layers (like done in our proposed models) outperforms other approaches.

To the best of our knowledge, this is the first time that therapy-related metadata, such as the clinic, the level of care, and the population, are taken into consideration for the task of psychotherapy quality assessment. Our results indicate that the incorporation of such information can yield significant performance improvements, thus suggesting that, depending on the specific use case and on data availability, several of those variables can be of high value for real-world systems deployed in clinical settings.

## Automatic behavioral coding

There has been an increasing interest in developing systems for automatic psychotherapy evaluation over the last few years, focusing on both acoustic (e.g., [11, 57]) and textual information. Depending on the domain, coding procedures may be applied at different resolutions, i.e., at the utterance (e.g., [58, 59]) or at the session level. We are limiting this short overview to language-based approaches for obtaining session-level codes, since this is the focus of the current article.

Early works in the field employed n-gram models [60, 61] and domain-specific semantic features [62] coupled with maximum likelihood [61], maximum entropy [12], and support vector machine [62] classifiers. Linguistic similarities between the therapist and the patient

have been also studied as a useful predictor of therapist behaviors [63, 64]. Deep learning techniques have opened up the way for more accurate and context-rich language modeling and better performance for the behavioral coding task [14, 65].

In the CBT domain, a large corpus of written posts from an online platform was used in [18], where the authors examined several deep learning approaches for CBT-related mental health concept understanding. However, online posts are typically much shorter than an actual CBT session and exhibit a more well-defined structure than a spontaneous conversational interaction. On the contrary, our model has been explicitly trained to evaluate the mental health provider's skills during verbal-based psychotherapy, utilizing a large set of recorded, real-world therapy sessions for training.

In our own previous work, we compared various linguistic features on a limited dataset of therapy transcriptions, both manually and automatically derived, and demonstrated that simple language representations, such as tf-idf features, can achieve competitive results for the task of automated cognitive therapy evaluation [16]. The same dataset was later utilized in conjunction with additional sessions from a different psychotherapy domain—namely, Motivational Interviewing (MI)—in a multi-task setting [66]. However, the dataset was selected to showcase a scenario focusing on the two extremes of the rating scale with mostly very low and very high CTRS values, whereas in this work we used a much larger and diverse—with respect to CTRS-based quality—set of CBT sessions. The tf-idf based approach was improved in [17] by enhancing the features with information again distilled from MI. However, this method assumes access to therapy sessions coded with an MI-based rating scheme at the utterance level in order to train and use an extra behavioral coding model and provide the extra information needed.

## Conclusion

In this work we introduced a model for quality assessment of psychotherapy sessions based on adapted BERT representations of therapy language use. The focus of the analysis was on the binary classification of CBT sessions with respect to the overall CTRS score, a metric commonly used in psychotherapy research to assess adherence to CBT skills. Two main architectures were proposed and compared. One was based on a single-task approach directly modeling the total CTRS as a binary output. The other took advantage of the definition of the total CTRS as the sum of 11 constituent scores, and was instead based on a multi-task approach where each score defined a task. Additionally, non-linguistic information was given to the models in the form of metadata variables modeling critical session and therapist characteristics. Experimental results showed that the best performance is achieved employing the multi-task network with metadata information.

Our experiments demonstrated the efficacy and validity of the proposed methods, but at the same time revealed areas for improvement and potential future research directions. We saw, for instance, that simple frequency-based linguistic features are still very relevant, achieving competitive results for the task of CBT quality assessment. Such results suggest that a combination of different linguistic representations and machine learning techniques may be beneficial in order to take advantage of both localized and contextualized patterns. At the same time, we believe that the performance of models based on contextualized representations, coupled with data-hungry deep learning classifiers, will keep improving once similar systems are introduced in clinical practice and therapy sessions are recorded systematically, thus increasing the size of available training sets.

In this study we only focused on language usage. However, human coders, by actually listening to the audio recording, have access to much richer information. This extra stream of

information can be potentially combined with linguistic representations, leading to increased predictive power of the systems used for CTRS prediction. For example, vocal synchrony and arousal could be beneficial when dealing with CTRS dimensions related to the therapist-patient relationship (e.g., understanding), while speech rate could be correlated with using time efficiently (CTRS dimension of pacing). Finally, it should be highlighted that even the linguistic information our models had access to is highly noisy since it comes from an automated transcription pipeline. Error propagation through the various modules of the pipeline, from speech activity detection and speaker clustering to speech recognition, unavoidably degrades the performance of text-based behavioral coding models. With improved speech technologies and employing a low-error transcription system, we expect the performance of CTRS prediction systems to get better in the future.

In any case, it is crucial that we all abide by certain ethical principles when producing, releasing, or using tools aiming at automated evaluation of psychotherapy. To that end, we proposed a set of ethical considerations and practical recommendations along four main axes: data privacy, prudent usage, error handling, and bias mitigation. Moving forward, we are hopeful that, under a proper framework, similar systems will be adopted in clinical practice, leading to more efficient training and supervision, improved quality of services, and, eventually, more positive clinical outcomes.

## Acknowledgments

Special thanks to the University of Utah Counseling Center and to the Philadelphia Department of Behavioral Health and Intellectual Disability Services for their contribution to this work.

## Author Contributions

**Conceptualization:** Nikolaos Flemotomos, Victor R. Martinez, Torrey A. Creed, David C. Atkins, Shrikanth Narayanan.

**Data curation:** Nikolaos Flemotomos, Torrey A. Creed.

**Formal analysis:** Nikolaos Flemotomos.

**Funding acquisition:** Torrey A. Creed, David C. Atkins, Shrikanth Narayanan.

**Methodology:** Nikolaos Flemotomos, Victor R. Martinez, Zhuohao Chen.

**Resources:** Torrey A. Creed, David C. Atkins, Shrikanth Narayanan.

**Software:** Nikolaos Flemotomos, Victor R. Martinez.

**Supervision:** Torrey A. Creed, David C. Atkins, Shrikanth Narayanan.

**Visualization:** Nikolaos Flemotomos.

**Writing – original draft:** Nikolaos Flemotomos.

**Writing – review & editing:** Victor R. Martinez, Zhuohao Chen, Torrey A. Creed, David C. Atkins, Shrikanth Narayanan.

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
