## [Decision Letter · Decision Letter 0]

11 Aug 2021

PONE-D-21-06203

Automated Quality Assessment of Cognitive Behavioral Therapy Sessions Through Highly Contextualized Language Representations

PLOS ONE

Dear Authors,

Thank you for submitting your manuscript to PLOS ONE. After careful consideration, we feel that it has merit but does not fully meet PLOS ONE’s publication criteria as it currently stands. Therefore, we invite you to submit a revised version of the manuscript that addresses the points raised during the review process.

We look forward to receiving your revised manuscript.

Kind regards,

Marcel Pikhart

Academic Editor

PLOS ONE

Journal Requirements:

2. PLOS ONE has specific requirements for studies that are presenting a new method or tool as the primary focus (https://journals.plos.org/plosone/s/submission-guidelines#loc-methods-software-databases-and-tools.) One requirement is that the tool must meet the criterion for utility. Specifically, the tool must be of use to the community and must present a proven advantage over existing alternatives, where applicable. To that effect, please describe in further detail how your tool presents advantages over existing alternatives, drawing direct comparisons between the alternatives and your current tool.

3. Thank you for stating the following in the Competing Interests/Financial Disclosure * (delete as necessary) section:

“I have read the journal's policy and the authors of this manuscript have the following competing interests: David C. Atkins and Shrikanth Narayanan are co-founders with equity stake in a technology company, Lyssn.io, focused on tools to support training, supervision, and quality assurance of psychotherapy and counseling. Torrey A. Creed is an advisor with an equity stake in Lyssn.io. Shrikanth Narayanan is also Chief Scientist and co-founder with equity stake in Behavioral Signal Technologies, a company focused on creating technologies for emotional and behavioral machine intelligence. The remaining authors report no conflicts of interest.”

We note that you received funding from a commercial source: Lyssn.io & Behavioral Signal Technologies

“This work was supported by the National Institutes of Mental Health (NIMH), grant 504 number R56 MH118550. Special thanks to the University of Utah Counseling Center for 505 their contribution to this work. UCC data collection was supported by the National 506 Institute of Alcoholism and Alcohol Abuse (NIAAA), grant number R01 AA018673.”

“TAC, DCA, and SN received a grant from the National Institutes of Mental Health (NIMH; www.nimh.nih.gov) - grant number R56 MH118550.

DCA and SN received a grant from the National Institute of Alcoholism and Alcohol Abuse (NIAAA; www.niaaa.nih.gov) - grant number R01 AA018673.

TAC received funding from the Philadelphia Department of Behavioral Health and Intellectual Disability Services (DBHIDS).”

6. We note that you have referenced (Young JE) which has currently not yet been accepted for publication. Please remove this from your References and amend this to state in the body of your manuscript: (Young JE) et al. [Unpublished]”) as detailed online in our guide for authors http://journals.plos.org/plosone/s/submission-guidelines#loc-reference-style

Reviewers' comments:

Reviewer's Responses to Questions

**Comments to the Author**

1. Is the manuscript technically sound, and do the data support the conclusions?

Reviewer #1: Yes

Reviewer #2: Yes

2. Has the statistical analysis been performed appropriately and rigorously? 

Reviewer #1: Yes

Reviewer #2: N/A

3. Have the authors made all data underlying the findings in their manuscript fully available?

Reviewer #1: No

Reviewer #2: Yes

4. Is the manuscript presented in an intelligible fashion and written in standard English?

Reviewer #1: Yes

Reviewer #2: Yes

5. Review Comments to the Author

Reviewer #1: Summary

The authors propose a deep learning solution to predict the level (high or low) of the cumulative cognitive therapy rating scale (CTRS) score evaluating the cognitive behaviroal therapy (CBT)-based psychotherapy sessions. The speech samples from a psychotherapy session are preprocessed to identify the utterances corresponding to the therapist and the patient/s. The utterances are transformed into sentence vectors using BERT-based language representations. The mapping between the utterance sequences and the associated rating level (high or low) is learned using recurrent neural networks comprising two approaches -- single task and multi-task. In addition, attention is used to learn the weightage of different score components on the final outcome. Therapy metadata variables have also been augmented to the model. In the single-task approach the the hidden state vectors from the recurrent networks are combined with attention weights to predict the level of CTRS score. In the multi-task approach individual models are trained for each component of the CTRS score and then the individual attention vectors are combined to predict the level of the cumulutaive CTRS score. The paper is well written and the extensive experiments and abltation studies establish the viability of the proposed approach. The proposed solution is a commendable addition to the toolkit for automating the evaluation of CBT sessions and has the promise of reducing the tedious task of manually evaluating audio samples from CBT sessions. The interpretability aspect of the proposed solution has promise in providing automatic, quality feedback to CBT trainees. The proposed solution makes technological advances compared to the earlier approaches that used traditional language features (frequency-based / handcrafted), by using the state-of-the-art transformer models. The results have been adequately discussed.

Major Comments:

1) Authors do not address the possible snowballing error from speech to text, speaker identification, text processing etc steps (ref to WER mentioned in lines 76-80) in the discussion.

2) "Data Sets" Section: It may be good to highlight which of the 11 CTRS codes are amenable for language-based representations and which ones are not. For example, it is not clear if “pt (pacing and efficient use of time)” could be represented using text based language features.

3) Table 3 does not present cross model performance scores. Please include this and discuss the differences in the performance of different models.

4) Authors used the "next sentence prediction" task to assess the performance of the BERT adaptations. Please motivate why this task is useful in the context of CBT session scoring as opposed to, say, performance on the "masked language modeling" task.

5) Comparison with the previous attempts: There are recent works from their group (links given below and also cited by the authors) that have used different feature sets (frequency based) and GloVe embeddings for the same task. Their aim and motivation was also to automate the evaluation process. No comparison of results is done with these attempts. It appears that the previous attempts yielded similar or better results compared to the proposed solutions. It would be nice to see discussion on why models based on frequency based features worked better as compared to the proposed model and the pros and cons thereof.

https://arxiv.org/pdf/2005.07809.pdf (Oct 2020)

https://www.researchgate.net/publication/327388874_Language_Features_for_Automated_Evaluation_of_Cognitive_Behavior_Psychotherapy_Sessions (Sept 2018)

6) Individual CTRS code evaluation: The previous attempts (mentioned above) experimented with predicting individual CTRS codes and evaluated the set of features that worked better for a particular CTRS code prediction. The approach taken in the current proposal is to focus on the cumulative score. Again, some discussion would be really nice on the design choice, especially given that the multi-task approach segregrates models for each CTRS code. Also, in the multi-task approach, instead of a loss function that considers cumulative CTRS score at the final stage, one could have broken this down into 11 models (one for each CTRS score). The loss function could have been simply based on the ground-truth value of the individual CTRS score for each model and a joint optimization for both cumulative score prediction as well as individual score predictions could have been desgined.

7) Reproducibility: Architectural details (layers, sizes, various parameters, hyper parameters) are not included in the paper. Also some sample uttarances from the sessions may also be included. These may be included for completeness and enabling reproducibility.

Minor Editorial suggestions:

1) Typo: line 67: that -> than

2) Typo: line 194: hight -> high

3) Ref 24: Flemotomos et al (2020) is incomplete

Reviewer #2: It is an interesting study and more relevant to the latest advancement in the field of psychology. Following are the suggestions to authors

1- topic of the study is interesting but not sample specific. Which kind of CBT sessions are evaluated, how the sessions similarity and differences were maintained?

2- Abstract: It is just theoretical, provide structured abstract with proper, background, objectives, method, results and conclusion. if you want to provide un structure abstract then must provide, objective, method, results and conclusion.

3- At the end of the introduction provide rationale of the study. data set shift to method section.

4- Provide sound literature which is supporting the research model

5- Method section covers sufficient details but core details of the method section are missing

6- Results section should be the separate heading

7- Discussion is fine.

8- Once cross check the references

6. PLOS authors have the option to publish the peer review history of their article (what does this mean?). If published, this will include your full peer review and any attached files.

Reviewer #1: **Yes: **Raju Surampudi Bapi

Reviewer #2: **Yes: **Qasir Abbas PhD

---

## [Author Response · Author response to Decision Letter 0]

29 Sep 2021

** ACADEMIC EDITOR **

While preparing the manuscript, we followed the PLOS ONE LaTeX template. Additionally, we read again in detail the style templates you kindly provided, and we made sure to follow the guidelines mentioned making the necessary changes (e.g., having all the headings in small letters apart from the first letter of the heading, properly updating the format of author affiliations, etc.).

2. PLOS ONE has specific requirements for studies that are presenting a new method or tool as the primary focus. One requirement is that the tool must meet the criterion for utility. Specifically, the tool must be of use to the community and must present a proven advantage over existing alternatives, where applicable. To that effect, please describe in further detail how your tool presents advantages over existing alternatives, drawing direct comparisons between the alternatives and your current tool.

In this revised version of the manuscript, we have tried to make clearer the advantages of our method over alternatives throughout the entire paper, and especially in the ‘Prior work’ section. Additionally, we have added, for completeness, a comparison with the tf-idf approach in the ‘Results and discussion – CTRS prediction’ section. 

3. Thank you for stating the following in the Competing Interests/Financial Disclosure * (delete as necessary) section:

“I have read the journal's policy and the authors of this manuscript have the following competing interests: David C. Atkins and Shrikanth Narayanan are co-founders with equity stake in a technology company, Lyssn.io, focused on tools to support training, supervision, and quality assurance of psychotherapy and counseling. Torrey A. Creed is an advisor with an equity stake in Lyssn.io. Shrikanth Narayanan is also Chief Scientist and co-founder with equity stake in Behavioral Signal Technologies, a company focused on creating technologies for emotional and behavioral machine intelligence. The remaining authors report no conflicts of interest.”

We note that you received funding from a commercial source: Lyssn.io & Behavioral Signal Technologies

While Shrikanth Narayanan, Torrey Creed and Daivid Atkins do report a conflict of interest with Lyssn.io and Behavioral Signal Technologies, those sources did not provide funding for the particular study in any way. We have included our updated Competing Interests statement in our cover letter and we also include it here:

“The authors have read the journal's policy and the authors of this study have the following competing interests to declare: David C. Atkins and Shrikanth Narayanan are co-founders with equity stake in a technology company, Lyssn.io, focused on tools to support training, supervision, and quality assurance of psychotherapy and counseling. Torrey A. Creed is an advisor with an equity stake in Lyssn.io. Shrikanth Narayanan is also Chief Scientist and co-founder with equity stake in Behavioral Signal Technologies, a company focused on creating technologies for emotional and behavioral machine intelligence. This does not alter our adherence to PLOS ONE policies on sharing data and materials. The remaining authors report no conflicts of interest. There are no patents, products in development or marketed products associated with this research to declare.”

“This work was supported by the National Institutes of Mental Health (NIMH), grant 504 number R56 MH118550. Special thanks to the University of Utah Counseling Center for 505 their contribution to this work. UCC data collection was supported by the National 506 Institute of Alcoholism and Alcohol Abuse (NIAAA), grant number R01 AA018673.”

“TAC, DCA, and SN received a grant from the National Institutes of Mental Health (NIMH; www.nimh.nih.gov) - grant number R56 MH118550.

DCA and SN received a grant from the National Institute of Alcoholism and Alcohol Abuse (NIAAA; www.niaaa.nih.gov) - grant number R01 AA018673.

TAC received funding from the Philadelphia Department of Behavioral Health and Intellectual Disability Services (DBHIDS).”

We have, now, updated the Acknowledgments section as follows:

“Special thanks to the University of Utah Counseling Center and to the Philadelphia Department of Behavioral Health and Intellectual Disability Services for their contribution to this work.”

Please use the Funding Statement you currently have on file, unaltered.

We added our ethics statement as the last paragraph of the ‘Datasets’ subsection under the new ‘Materials and methods’ section.

6. We note that you have referenced (Young JE) which has currently not yet been accepted for publication. Please remove this from your References and amend this to state in the body of your manuscript: (Young JE) et al. [Unpublished]”) as detailed online in our guide for authors http://journals.plos.org/plosone/s/submission-guidelines#loc-reference-style

We have now fixed this issue, by removing this item from the reference list and mentioning it in the body of the manuscript as follows:

“(the CTRS manual is available from \\url{https://www.academyofct.org/general/custom.asp? page=CTRSCRRS} [Young JE, Beck JS; Unpublished])”

We have made the following changes in the reference list:

We added pages for (Xiao et al., 2014)

We fixed the article volume/number for (Atkins et al., 2014)

We updated (Flemotomos et al., 2020) which is now published in Behavior Research Methods.

We updated (Chen et al., 2021) which is now accepted for publication and will appear in Proceedings of EMBC.

Per the journal’s guidelines, we added DOI for (Gibson et al., 2019), since the traditional volume/pages are not available

We added the following articles in the reference list to support our methods and arguments: (Yu et al., 2021), (Sun et al., 2019), (Zhou et al., 2016), (Britz et al., 2017), (Li et al., 2020), (Randall et al., 2017) 

Following the recommended citation from Interspeech, we updated all the references which appeared in Interspeech to “In: Proc. Interspeech” from “In: Proc. Annual Conference of the International Speech Communication Association (Interspeech)”

** REVIEWER 1 **

1. Authors do not address the possible snowballing error from speech to text, speaker identification, text processing etc steps (ref to WER mentioned in lines 76-80) in the discussion.

This is, indeed, an important aspect of our work. We have shown in this study that even using a transcription pipeline of a relatively high error rate, we can get good classification results with respect to the total CTRS. The reported error (WER) is inflated because of frequent fillers and short utterances that our models potentially fail to capture. Of course, there is great room for improvement in all those pre-processing steps and the errors inevitably propagate to the downstream task of CTRS-based classification. We have, now, added relevant discussions both in the ‘Datasets’ section and in the ‘Conclusion’. 

2. "Data Sets" Section: It may be good to highlight which of the 11 CTRS codes are amenable for language-based representations and which ones are not. For example, it is not clear if “pt (pacing and efficient use of time)” could be represented using text based language features.

Even though all the CTRS codes are related to skills which, at some extent, can be expressed verbally, it is true that some of them are better represented by linguistic features than others. Combining linguistic and non-linguistic (i.e., audio) representations is an interesting direction of future research. We have added relevant discussions both in the ‘Datasets’ section and in the ‘Conclusion’.

3. Table 3 does not present cross model performance scores. Please include this and discuss the differences in the performance of different models.

We have, now, included those scores (in the table now labeled as Table 4) with new discussion of the results underneath. 

4. Authors used the "next sentence prediction" task to assess the performance of the BERT adaptations. Please motivate why this task is useful in the context of CBT session scoring as opposed to, say, performance on the "masked language modeling" task.

We have added an extended discussion on why the next sentence prediction task is important in the context of psychotherapy in the section now called ‘BERT model performance’.

5. Comparison with the previous attempts: There are recent works from their group (links given below and also cited by the authors) that have used different feature sets (frequency based) and GloVe embeddings for the same task. Their aim and motivation was also to automate the evaluation process. No comparison of results is done with these attempts. It appears that the previous attempts yielded similar or better results compared to the proposed solutions. It would be nice to see discussion on why models based on frequency based features worked better as compared to the proposed model and the pros and cons thereof.

https://arxiv.org/pdf/2005.07809.pdf (Oct 2020)

https://www.researchgate.net/publication/327388874_Language_Features_for_Automated_Evaluation_of_Cognitive_Behavior_Psychotherapy_Sessions (Sept 2018)

The results reported in the linked papers are not directly comparable to the results in our current manuscript, since in both those papers the models are trained and evaluated on a much smaller dataset. Especially for the one from Sept 2018, most of the results are reported for manually transcribed therapy sessions which were selected to showcase a scenario with very low vs. very high CTRS scores. Here, we have a much larger dataset with sessions covering the entire spectrum of CTRS. For completeness, we have now added results with the tf-idf approach, which gave the best results in this paper, together with related discussion. The method proposed in Oct 2020, though improves the traditional tf-idf approach, assumes that we have available another trained system (or available data to train one) which codes according to a coding scheme (MISC) used in a different type of psychotherapy (namely, Motivational Interviewing – MI) and not in CBT – as we now mention in our manuscript. Even though the combination of MI-based and CBT-based techniques from a computational perspective is an exciting research area (with promising results in behavioral coding, as shown both in the linked paper from Oct 2020 and in (Gibson et al., 2019)), we believe that it goes beyond the scope of our current study which focuses on CBT-specific techniques and models.

6. Individual CTRS code evaluation: The previous attempts (mentioned above) experimented with predicting individual CTRS codes and evaluated the set of features that worked better for a particular CTRS code prediction. The approach taken in the current proposal is to focus on the cumulative score. Again, some discussion would be really nice on the design choice, especially given that the multi-task approach segregrates models for each CTRS code. Also, in the multi-task approach, instead of a loss function that considers cumulative CTRS score at the final stage, one could have broken this down into 11 models (one for each CTRS score). The loss function could have been simply based on the ground-truth value of the individual CTRS score for each model and a joint optimization for both cumulative score prediction as well as individual score predictions could have been desgined.

The cumulative score (which we call the ‘total CTRS’ or `overall CTRS’) is the one taken into account in clinical settings, and this is why we have focused on this in the current study. Even though we do propose the multi-task approach which deals with each code ‘independently’, we do so for improved interpretability, while our focus is still to train a single model which outputs the total CTRS. We have tried to make our goal and motivation clearer in this revised version of the manuscript. 

As for the loss function of the multi-task approach, this is indeed based on the ground-truth value of the individual CTRS scores, as suggested, since the loss function of the system is simply the unweighted sum of the individual losses. We highlighted this fact in this revised version.

7. Reproducibility: Architectural details (layers, sizes, various parameters, hyper parameters) are not included in the paper. Also some sample uttarances from the sessions may also be included. These may be included for completeness and enabling reproducibility.

We have, now, added any details which were missing from the original manuscript in the renamed ‘Experimental setup’ section. Additionally, we have added sample utterances in the ‘Datasets’ section with some relevant discussion.

Minor Editorial suggestions:

1) Typo: line 67: that -> than

2) Typo: line 194: hight -> high

3) Ref 24: Flemotomos et al (2020) is incomplete

Thank you for pointing out those issues, which we have now fixed.

** REVIEWER 2 **

1. topic of the study is interesting but not sample specific. Which kind of CBT sessions are evaluated, how the sessions similarity and differences were maintained?

The CBT sessions included in this sample are intentionally heterogeneous in terms of population, presenting problem, and setting in order to support the generalizability of the findings. More specifically, the sessions were collected through the Beck Community Initiative (BCI) at the University of Pennsylvania, which is a large public-academic partnership that has implemented CBT in more than 75 diverse community mental health contexts. Rather than limiting inclusion to sessions with patients with a specific disorder, the BCI uses a transdiagnostic CBT approach that is appropriate across the broad spectrum of presenting problems and populations seen in community care. The CTRS is the established gold-standard measure of competence in CBT and is the most commonly used measure to evaluate transdiagnostic CBT skills. 

We have now included those details in the “Datasets” section.

2. Abstract: It is just theoretical, provide structured abstract with proper, background, objectives, method, results and conclusion. if you want to provide un structure abstract then must provide, objective, method, results and conclusion.

We wanted the abstract to be general enough since the readership of the journal includes people from various backgrounds. We have updated it in this revised version of the manuscript and, even though we do not provide a ‘structured’ abstract with headings (since, to the best of our knowledge, this is not a requirement in PLOS ONE), we did add details about our method and results which were absent.

3. At the end of the introduction provide rationale of the study. data set shift to method section.

We added a paragraph in the Introduction that we believe makes our rationale and motivation clearer. We have, also, included ‘Datasets’ as a subsection in the newly named ‘Materials and methods’ section.

4. Provide sound literature which is supporting the research model

In the ‘Materials and methods’ section (single-task and multi-task approach), we added pointers to articles we already had in the reference list and we also added a few articles that support our design decisions (e.g., BERT adaptation, multi-task learning, inclusion of metadata information)

5. Method section covers sufficient details but core details of the method section are missing

We have, now, updated the format of the manuscript, adding several details about our method in the ‘Materials and methods’ section, in both the ‘Single-task approach’/’Multi-task approach’ and the ‘Experimental setup’ subsections.

6. Results section should be the separate heading

We have moved the results, together with the relevant discussions, to a section called ‘Results and discussion’, as recommended in the PLOS ONE style template.

7. Discussion is fine.

8. Once cross check the references

We checked once again the reference list and we made a few updates. Specifically, as we also mention in our response to the Academic Editor:

We added pages for (Xiao et al., 2014)

We fixed the article volume/number for (Atkins et al., 2014)

We updated (Flemotomos et al., 2020) which is now published in Behavior Research Methods.

We updated (Chen et al., 2021) which is now accepted for publication and will appear in Proceedings of EMBC.

Per the journal’s guidelines, we added DOI for (Gibson et al., 2019), since the traditional volume/pages are not available

We added the following articles in the reference list to support our methods and arguments: (Yu et al., 2021), (Sun et al., 2019), (Zhou et al., 2016), (Britz et al., 2017), (Li et al., 2020), (Randall et al., 2017) 

Following the recommended citation from Interspeech, we updated all the references which appeared in Interspeech to “In: Proc. Interspeech” from “In: Proc. Annual Conference of the International Speech Communication Association (Interspeech)”

---

## [Editor Report · Decision Letter 1]

4 Oct 2021

Automated quality assessment of cognitive behavioral therapy sessions through highly contextualized language representations

PONE-D-21-06203R1

Dear Authors,

We’re pleased to inform you that your manuscript has been judged scientifically suitable for publication and will be formally accepted for publication once it meets all outstanding technical requirements.

Kind regards,

Marcel Pikhart

Academic Editor

PLOS ONE
---

## [Editor Report · Acceptance letter]

14 Oct 2021

PONE-D-21-06203R1 

Automated quality assessment of cognitive behavioral therapy sessions through highly contextualized language representations 

Dear Dr. Flemotomos:

I'm pleased to inform you that your manuscript has been deemed suitable for publication in PLOS ONE. Congratulations! Your manuscript is now with our production department. 

Kind regards, 

on behalf of

Dr. Marcel Pikhart 

Academic Editor

PLOS ONE